# An Experimental Study on the Reduction Behavior of Dust Generated from Electric Arc Furnace

**Mengxu Zhang [1], Jianli Li [1,2,3,\*]** **, Qiang Zeng [1] and Qiqiang Mou [1]**

[1]   The State Key Laboratory of Refractories and Metallurgy, Wuhan University of Science and Technology, Wuhan 430081, China

[2]   Hubei Provincial Key Laboratory for New Processes of Ironmaking and Steelmaking, Wuhan University of Science and Technology, Wuhan 430081, China

[3]   Key Laboratory for Ferrous Metallurgy and Resources Utilization of Ministry of Education, Wuhan University of Science and Technology, Wuhan 430081, China

\*   Correspondence: jli@wust.edu.cn

**Featured Application: This work expects to propose a direct-recycling process on basis of the mixture of EAFD powder and reductant (carbon or ferrosilicon) loaded into an electric arc furnace, in order to recycle Zn and Fe in the form of dust and molten steel. The results would provide the theoretical reference.**

**Abstract:** To improve the utilization value of electric arc furnace dust (EAFD) containing zinc, the reduction behavior of non-agglomerate dust was investigated with carbon and ferrosilicon in an induction furnace. The experimental results show that when the temperature increases, the zinc evaporation rate increases. When the reducing agent is carbon, zinc evaporation mainly occurs in the range of 900–1100 °C. When the reducing agent is ferrosilicon, zinc begins to evaporate at 800 °C, but the zinc evaporation rate is 90.47% at 1200 °C and lower than 99.80% with carbon used as a reducing agent at 1200 °C. For the carbon reduction, the iron metallization rate increases with a rise in the temperature. When the reducing agent is ferrosilicon, with an increase in temperature, the metallization rate first increases, then decreases, and finally, increases, which is mainly due to the reaction between the metallic iron and ZnO. In addition, the residual zinc in the EAFD is mainly dispersed in the form of a spinel solution near the metallic phase.

**Keywords:** EAFD; reduction; Zinc; FeO; evaporation; metallization

---

## 1. Introduction

Electric arc furnace dust (EAFD) is one of the by-products of electric furnace steelmaking, and its production accounts for 1~2% of the charging amount of steel scrap and other metal materials [1,2]. Because heavy metal elements like lead, chromium, and cadmium are contained in EAFD, these kinds of by-products are classified as toxic waste by many countries [3]. However, dust that contains iron, lead, zinc, and chromium can be used as a secondary resource. Therefore, the recovery of these resources plays an important role in environmental protection [4–7]. However, the zinc content of EAFD is generally <25 wt%, which does not meet the raw material requirements for the non-ferrous industry. Thus, a part of the EAFD can only be stacked or landfilled as disposal [8]. In order to reduce pollutant emissions and promote ecological progress, the environmental protection laws of the People's Republic of China stipulate that the taxes for steelmaking slag, steelmaking dust, coal ash, and other solid waste are 25 yuan per ton. However, if the solid waste is comprehensively utilized and meets the standards of the national and local environmental protection laws, the corresponding tax can be

temporarily exempted. The output of the EAFD increased rapidly with increases in the output of EAF steelmaking [9,10]. Therefore, the treatment and comprehensive utilization of EAFD is gradually focused, and a variety of studies have been carried out.

At present, the Waelz kiln process, the rotary hearth furnace process (RHF), and the OxyCup process are the main processes used to treat steelmaking dust [11]. Although these processes can recycle the dust and sludge of iron and steel enterprises to some extent, they all have shortcomings. For example, the Waelz kiln process has higher requirements for raw materials, higher energy consumption, unstable production and operation, and a limited processing scale [12]. The main product of the RHF is metallized pellets, but its product has a high sulfur content, and the process has a low production efficiency and a large investment [13]. The OxyCup process's product is molten iron, and its requirements for raw materials are relatively broad. However, its equipment operation cycle is short, and its maintenance work is difficult [11]. Furthermore, a homogenous mixture of EAF dust, reductant, and flux needs to be initially prepared in pellet form during the above processes [14]. In view of their shortcomings, it is an urgent issue to propose an eco-friendly practical process to recycle and utilize the valuable elements in the dust of iron and steel enterprises on the basis of environmental protection. Xu [15] and Zhang [16] mixed electric furnace dust with a reducing agent, without lumping or pelletizing, and then charged the mixture from the EAF directly into bags, which not only reduced the dust production by 40% and increased the zinc content in the dust to 29.7%, but also directly recycled valuable elements, such as iron, in the dust. Normal smelting is not affected by this process, but foaming slag occurs, while the zinc content of secondary dust without charging pellets is only 21% zinc.

In order to further understand the behavior of zinc and iron in EAFD at different temperatures and provide theoretical reference for the recycling of EAFD containing zinc, the evaporation behavior of zinc and the metallization ratio of iron during EAFD reduction via carbon and ferrosilicon are investigated in this work.

## 2. Experimental

The EAFD was obtained from an electric arc furnace (Xingtai Iron & Steel Corp. LTD, Xingtai, China). As shown in Table 1, $Fe_2O_3$ and ZnO were the main components of the dust, together comprising 67.09 wt% of it. In the dust, there was, simultaneously, CaO, MgO, $SiO_2$, $Cr_2O_3$, MnO, and a small amount of $Al_2O_3$. Fe and Zn mainly existed in $ZnFe_2O_4$, $Fe_3O_4$, and ZnO, which were also the main phases of EAFD (Figure 1). As shown in Figure 2, EAFD mainly consisted of small particles, which were less than 2 μm. Ferrosilicon and carbon powder were used as reducing agents in the experiments. The ferrosilicon contained 74.64% Si and 24.00% Fe, and the content of C in the carbon powder was >99%. Both were ground into powder for use.

$Fe_2O_3$, ZnO, and MnO in the EAFD can be reduced to Fe, Zn, and Mn via the reducing agents. Based on Table 1, it can be calculated that when 100 g EAFD was reduced by silicon or carbon, the required amounts of silicon and carbon were 15.63 g and 14.72 g, respectively. Since the ferrosilicon alloy was the carrier of silicon, its silicon content was 74.64%. Therefore, the actual amount of ferrosilicon alloy for 100 g dust was 20.92 g. In order to completely reduce the sample, the amount of the dispensed reducing agent should be slightly excessive. Therefore, the final amount of the ferrosilicon alloy and carbon powder was set to 22 wt% and 16 wt%, respectively.

**Table 1.** The composition of electric arc furnace dust (EAFD) in wt%.

| Component | CaO | MgO | $SiO_2$ | $Al_2O_3$ | $Cr_2O_3$ | $Fe_2O_3$ | ZnO | MnO | KCl | $CaF_2$ |
|---|---|---|---|---|---|---|---|---|---|---|
| Content | 6.80 | 5.83 | 5.67 | 0.35 | 5.02 | 40.30 | 26.79 | 2.27 | 0.88 | 1.6 |

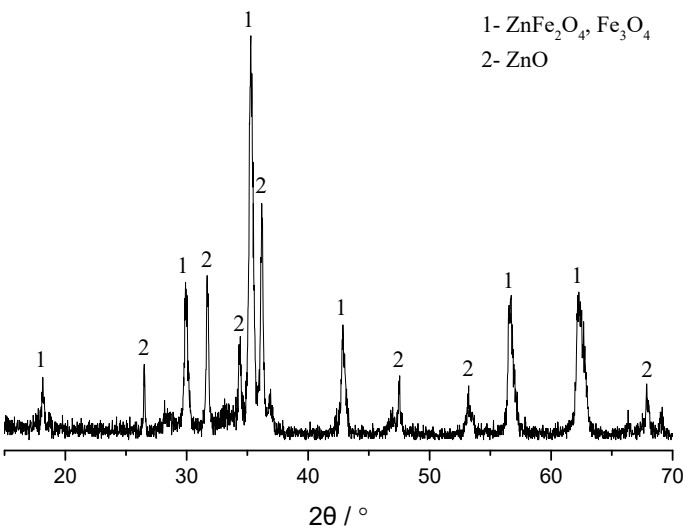

**Figure 1.** XRD diffraction pattern of EAFD.

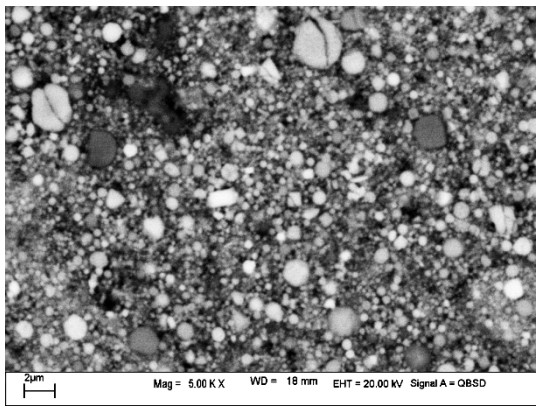

**Figure 2.** The microstructure of EAFD.

This experiment includes two series, referred to as A and B. A total of 50 g EAFD in series A was mixed with 8 g carbon powder, while 50 g EAFD in series B was mixed with 11 g ferrosilicon alloy powder. The uniform mixing samples were charged into the corundum crucible (inner diameter 32 mm, height 65 mm, and wall thickness 1 mm) with a corundum cover. The crucible used in this experiment was completely dried in a muffle furnace under 200 °C for 3 h in order to eliminate the influence of water. The crucible containing the materials was loaded into an induction furnace (60 kW, 3 kHz) and kept at different temperatures for 30 min. Six temperatures were selected: 700 °C, 800 °C, 900 °C, 1000 °C, 1100 °C, and 1200 °C. The numbers of the experiments are shown in Table 2. After keeping a constant temperature for the predetermined time, the samples were naturally cooled to room-temperature in the furnace.

**Table 2.** Numbers of the experiments.

| Category | 700 °C | 800 °C | 900 °C | 1000 °C | 1100 °C | 1200 °C |
|----------|--------|--------|--------|---------|---------|---------|
| A | A1 | A2 | A3 | A4 | A5 | A6 |
| B | B1 | B2 | B3 | B4 | B5 | B6 |

The characteristics of the samples were analyzed through X-ray fluorescence (XRF-1800, Shimadzu, Kyoto, Japan), X-ray diffraction (XRD, X Pert Pro MPD, PANalytica, Almelo, Netherlands), and a scanning electron microscope equipped with an energy dispersive spectrometer (SEM-EDS, NanoSEM400, FEI, Hillsboro, OR, USA). The total iron, metallic iron, ferrous iron, and zinc content of

the samples were determined by the reduction method of titanium trichloride, the volumetric method of ferric chloride and sodium acetate, the volumetric method of potassium dichromate, and flame atomic absorption spectrometry, respectively. The evaporation rate of zinc ($R_{Zn}$) and the metallization rate of iron ($R_{Fe}$) are defined as Equations (1) and (2), respectively. It was assumed that the CaO quality in the samples remained constant during the heating process and that the quality of the samples after the reaction can be obtained according to the CaO content in the samples:

$$R_{Zn} = \frac{T_{Zn} - L_{Zn}}{T_{Zn}} \times 100\% \tag{1}$$

$$R_{Fe} = \frac{M_{Fe}}{T_{Fe}} \times 100\% \tag{2}$$

where $T_{Zn}$ is the total mass of zinc in the sample before the reaction. $L_{Zn}$ refers to the mass of the remaining zinc in the sample after reaction, $T_{Fe}$ represents the total iron content in the sample before reaction, and $M_{Fe}$ is the amount of metal iron in the sample after the reaction (all in grams).

## 3. Results

The chemical compositions of the samples after treatment are summarized in Table 3. When the temperature rose, the CaO, $SiO_2$, MgO, $Al_2O_3$, $Cr_2O_3$, and metallic iron content gradually increased. The FeO content increased first and then decreased, while the elemental zinc content gradually decreased in both series A and B. The residual zinc content was 0.07 wt% and 2.05 wt%, the metallic iron content was 37.00 wt% and 24.20 wt%, and the basicity of the sintered slags (CaO%/SiO$_2$%) was 1.09 and 0.20 in samples A6 and B6.

**Table 3.** The chemical composition of the samples after the reaction (in wt%).

| Sample | CaO | SiO$_2$ | MgO | Al$_2$O$_3$ | Cr$_2$O$_3$ | Fe$^{2+}$ | MFe | Zn |
|--------|------|---------|------|-------------|-------------|-----------|-------|-------|
| A1 | 6.76 | 5.05 | 5.23 | 0.34 | 4.53 | — | 0.15 | 21.10 |
| A2 | 6.80 | 5.40 | 5.37 | 0.34 | 4.61 | 6.75 | 0.65 | 19.10 |
| A3 | 7.10 | 5.37 | 5.31 | 0.32 | 4.59 | 15.80 | 0.93 | 18.90 |
| A4 | 8.10 | 6.91 | 7.12 | 0.43 | 6.60 | 11.20 | 23.10 | 14.10 |
| A5 | 10.10 | 9.22 | 10.10 | 0.61 | 7.25 | 11.20 | 35.10 | 0.31 |
| A6 | 9.90 | 9.10 | 9.99 | 0.59 | 7.37 | 9.10 | 37.00 | 0.07 |
| B1 | 6.20 | 29.30 | 4.85 | 0.62 | 4.13 | 16.90 | 5.60 | 17.60 |
| B2 | 6.50 | 30.10 | 5.18 | 0.64 | 4.49 | 14.20 | 16.20 | 12.00 |
| B3 | 6.90 | 32.80 | 5.40 | 0.70 | 5.26 | 15.60 | 15.50 | 10.60 |
| B4 | 6.90 | 33.70 | 5.41 | 0.72 | 4.63 | 20.30 | 13.80 | 8.75 |
| B5 | 6.90 | 34.70 | 5.67 | 0.74 | 4.81 | 14.10 | 19.00 | 6.10 |
| B6 | 7.20 | 35.60 | 5.96 | 0.98 | 4.96 | 16.80 | 24.20 | 2.05 |

### 3.1. Reduction and Evaporation Rates of Zinc

As can be seen from Table 3 and Figure 3, the evaporation rates of zinc in both series increased with a rise in temperature. When the reducing agent was carbon powder, the evaporation rate of zinc was 3.30% and 99.11% at 900 °C and 1100 °C, respectively. This suggests that the evaporation of zinc mainly occurs between 900–1100 °C, and the evaporation process can be divided into three stages. In this way, <900 °C is the dormant period, 900–1100 °C is the evaporation period, and >1100 °C is the final evaporation period. When ferrosilicon was used as the reducing agent, the evaporation of zinc increased gradually with an increase in temperature. The evaporation rate of zinc was 35.26% at 800 °C and only 90.47% at 1200 °C. Compared to the carbon powder, the temperature range of zinc evaporation is larger, and the evaporation rate is relatively smaller.

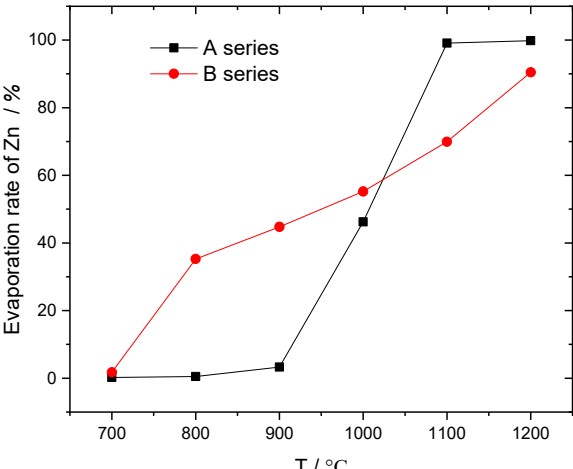

**Figure 3.** The effect of the reducing agent on the reduction and evaporation of zinc.

### 3.2. Metallization Ratio of Iron

As shown in Figure 4, the metallization rate of the iron increased somewhat in the samples of series A and B with an increase in temperature. When the carbon powder was used as the reducing agent, the metallization rate of sample A3 was only 3.63%, while that of sample A5 was as high as 77.12%, which is slightly lower than that of sample A6 (78.67%). Hence, the metallization percentage of the iron-changing trend can be divided into three stages, similar to the evaporation process of zinc: <900 °C is the dormant period, 900–1100 °C is the metallization occurrence period, and >1100 °C is the final metallization period. Figure 4 demonstrates that metallization mainly occurs between 900 °C and 1100 °C.

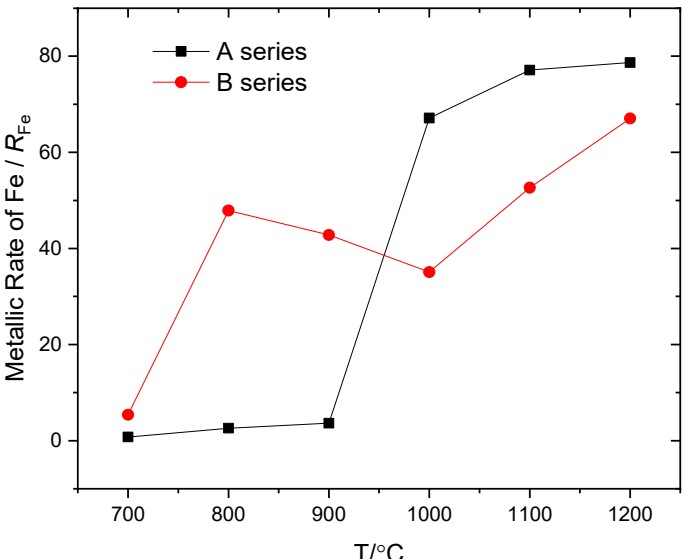

**Figure 4.** The effect of the reducing agent on the percentage metallization of iron.

When the reductant was ferrosilicon, the metallization rate of the B series samples increased first, then decreased, and finally increased with an upturn in temperature. The metallization rate was as high as 47.88% at 800 °C, which was far higher than the 2.58% of A2. The metallization rate of the B4 samples reduced to 35.08%, significantly lower than the 67.12% of A4. The metallization rate of B6 increased to 67.04% again, which is lower than the 78.67% of A6.

### 3.3. Occurrence State of Zinc

The microstructure and chemical components obtained by SEM-EDS are shown in Figure 5 and Table 4, individually. The mineral phases of samples A6 and B6 measured by XRD are shown in Figure 6. When the reducing agent was carbon powder, the EAFD samples maintained a relatively intact state for their particles, but the interface between the particles became vague at 1000 °C. According to the analysis of EDS and XRD, samples in A6 were clearly divided into two phases. Phase 1 is composed of Fe and Cr metal phases, and Phase 2 comprises a $CaMgSiO_4$ phase formed by CaO, MgO, and $SiO_2$. The residual zinc in the solid samples is lower than the detection limit of SEM-EDS.

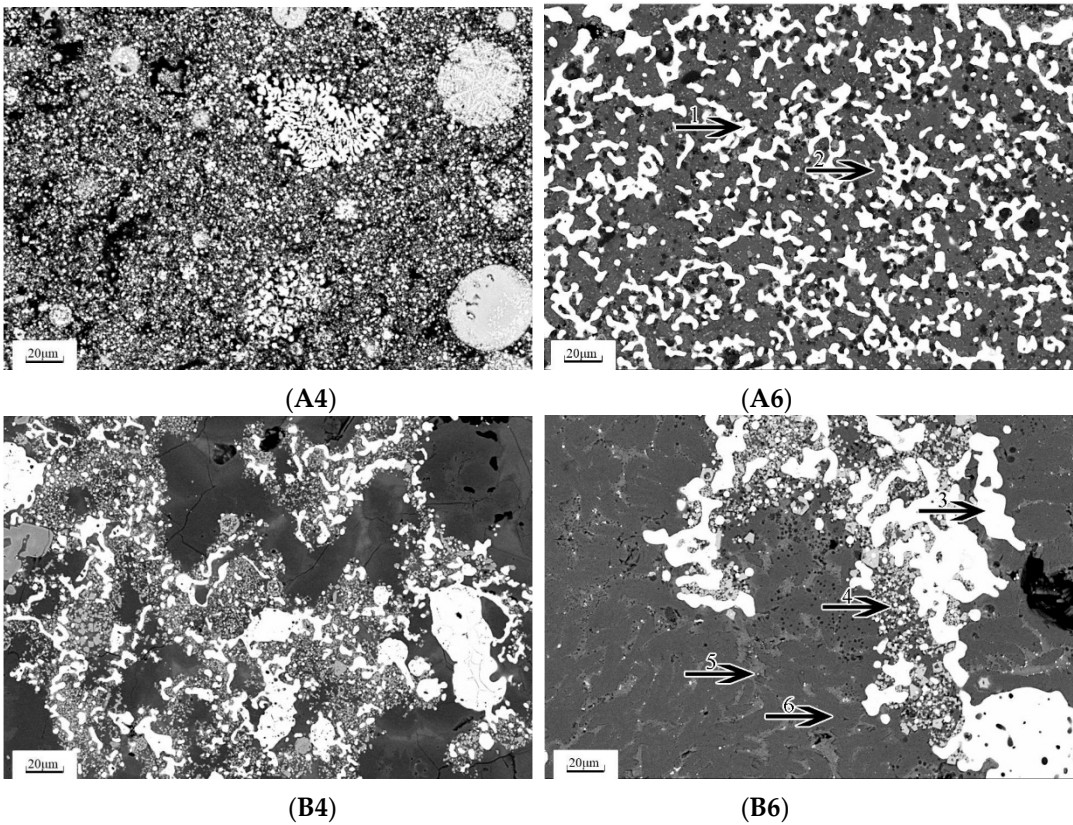

**Figure 5.** The microstructure of the sample after being reduced at 1000 °C and 1200 °C (A4 and A6 were reduced by carbon at 1000 °C and 1200 °C, respectively. B4 and B6 were reduced by ferrosilicon at 1000 °C and 1200 °C, respectively).

**Table 4.** Chemical components of the different phases in Figure 5 (in wt%).

| Phase | Ca | Si | Mg | Al | Cr | Fe | Zn | O |
|---|---|---|---|---|---|---|---|---|
| 1 | — | — | — | — | 2.62 | 95.23 | — | — |
| 2 | 36.93 | 18.39 | 7.25 | — | — | — | — | 36.26 |
| 3 | — | — | — | — | 0.35 | 98.48 | — | — |
| 4 | 8.09 | 3.84 | 5.91 | 1.94 | 31.50 | 8.59 | 4.67 | 31.14 |
| 5 | 5.88 | 40.31 | 2.28 | 1.72 | 0.78 | 2.42 | — | 43.58 |
| 6 | 14.74 | 29.40 | 11.09 | 1.61 | 1.25 | 1.94 | 0.33 | 37.20 |

When the reducing agent was ferrosilicon, the EAFD samples were melted, and the metal particles and the silicate phases were already formed at 1000 °C. The samples in B6 can be divided into four different phases. Phase 3 is a metal phase, composed of chromium and iron. Phase 4 is (Mg, Fe, Zn) $Cr_2O_4$ spinel phase, made up of MgO, FeO, ZnO, and $Cr_2O_3$. Phase 5 is an $SiO_2$ phase, and Phase 6 is

a $CaMgSi_2O_6$ diopside phase, composed of $CaO$, $MgO$, and $SiO_2$. The residual zinc in the solid phase is mainly dissolved into a spinel crystal and diopside phase.

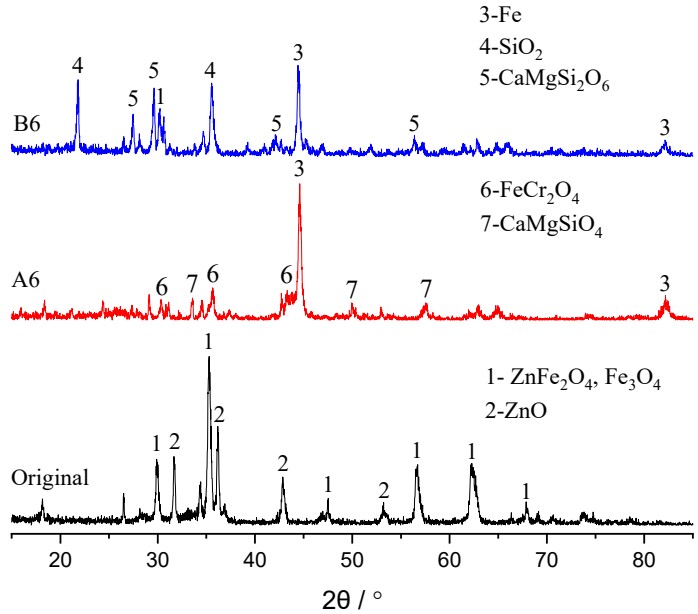

**Figure 6.** The mineral components of the samples obtained at 1200 °C.

## 4. Discussion

### 4.1. Characteristics of Reduction Processes with Carbon

As can be seen from Figure 1, the EAFD is composed of spherical particles, and the mixture of reducing agents and the EAFD is a dispersion system with good contact. According to the thermodynamics principle, the chemical reactions of EAFD containing zinc reduced by carbon powder are shown in Equations (3)–(11) [11,17,18], which comprise the direct reduction of solid carbon and the indirect reduction of CO:

$$ZnO(s) + C(s) = Zn(g) + CO(g) \quad \Delta G_1^\theta = 348480 - 286.1T \tag{3}$$

$$3Fe_2O_3(s) + C(s) = 2Fe_3O_4(s) + CO(g) \quad \Delta G_2^\theta = 4536909 - 224.37T \tag{4}$$

$$Fe_3O_4(s) + C(s) = 3FeO(s) + CO(g) \quad \Delta G_3^\theta = 207510 - 217.62T \tag{5}$$

$$FeO(s) + C(s) = Fe(s) + CO(g) \quad \Delta G_4^\theta = 158970 - 160.25T \tag{6}$$

$$ZnO(s) + CO(g) = Zn(g) + CO_2(g) \quad \Delta G_5^\theta = 178020 - 111.6T \tag{7}$$

$$3Fe_2O_3(s) + CO(g) = 2Fe_3O_4(s) + CO_2(g) \quad \Delta G_6^\theta = 35380 - 40.16T \tag{8}$$

$$Fe_3O_4(s) + CO(g) = 3FeO(s) + CO_2(g) \quad \Delta G_7^\theta = 71940 - 73.62T \tag{9}$$

$$FeO(s) + CO(g) = Fe(s) + CO_2(g) \quad \Delta G_8^\theta = -13160 + 17.21T \tag{10}$$

$$C(s) + CO_2(g) = 2CO(g) \quad \Delta G_9^\theta = 172130 - 177.46T \tag{11}$$

According to the above equations, the chemical reaction of EAFD containing zinc reduced by carbon powder is a complicated heterogeneous reaction process. As shown in Figure 7, when the temperature is less than 900 °C, a direct reduction reaction is the main reaction; the reactions are of the solid–solid type. When the temperature is higher than 900 °C, the reaction is principally indirect, belonging to a gas–solid reaction. The reaction rate of the multiphase reaction depends predominantly on the diffusion rate of the reaction components and the interfacial chemical reaction rate. Since the

solid reactants are in direct contact with each other before the start of the reaction, the reaction can be started primordially. However, a solid product layer (metal phase and slag phase) is formed between the dust particles and the solid carbon during the reduction procedure. Consequently, further progress of the reaction will depend on the diffusion of the reactants through the layer.

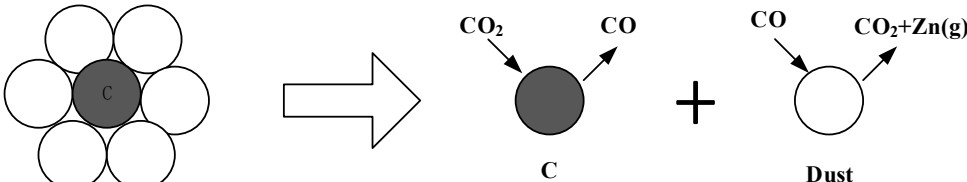

**Figure 7.** Schematic diagram of the reaction of EAFD containing zinc reduced by carbon.

Under the influence of the Boudouard reaction, Equation (11), the carbon-reduction process is transformed from a solid–solid reaction to a gas–solid phase reaction. The general steps for the corresponding process include the external diffusion of gas reactants and the products in the boundary layer on the solid surface, the internal diffusion of gas reactants, and products through the solid product layer, as well as the interfacial chemical reactions between gas and solid reactants. According to the microstructure of the dust samples (A5) reduced by carbon at 1100 °C, the particles in the sample still maintain a relatively independent shape and a certain number of pores between the particles, as shown in Figure 8. This provides a good condition for the diffusion of the reducing agent, CO, and reduction product, Zn (g).

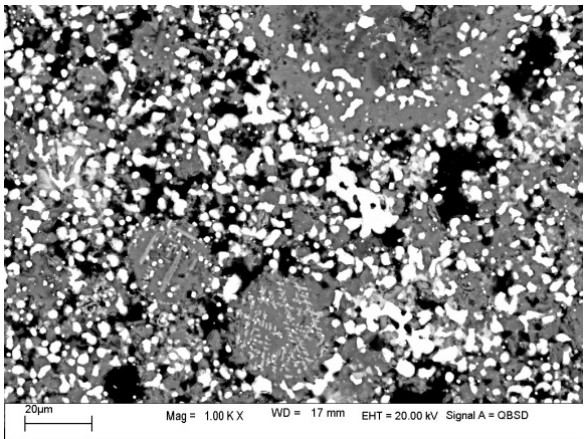

**Figure 8.** Microstructure of the reduced samples (A5) at 1100 °C.

Since the boiling point of zinc is 908.47 °C, the zinc vapor could be separated from EAFD and discharged as a gas. The evaporation rate of zinc is governed by an interfacial chemical reaction, the diffusion of the zinc vapor, and the temperature. In the low temperature stage (900 °C), the evaporation rate of zinc in the sample is very low because of the boiling point of zinc. With the mechanism transformation of the carbon-thermal reduction, the interface chemical reaction and the diffusion of CO and zinc (g) are improved when the temperature is higher than 908.47 °C. Therefore, the evaporation rate of zinc increased quickly in the temperature range of 900–1100 °C.

The alteration of the metallization rate can be explained in the following ways. In the low temperature stage (below 900 °C), the reduction of $Fe_2O_3$ mainly occurred in the sample because it was restricted by a solid–solid reaction, as in Equations (4) and (5). According to Table 3, it can be inferred that when the temperature increased from 700 °C to 900 °C, the $Fe^{2+}$ content in the samples increased from 0 to 15.80%, but the content of $M_{Fe}$ increased from 0.15 to 0.93%. When the temperature increased to above 900 °C, the interfacial chemical reaction transformed from a solid–solid reaction to

a gas–solid reaction, and good conditions were provided for the diffusion of CO in the samples. In this way, the metallization rate of iron increased rapidly in the range of 900–1100 °C.

## 4.2. Characteristics of Silicon Reduction

The reduction process accomplished by the ferrosilicon alloy is a solid–solid reaction, and the rate-controlling step of this reaction potentially includes the following possible situations: an interfacial chemical reaction, the diffusion of reactants through the product layer, and an interfacial chemical reaction and diffusion. The ferrosilicon reduction is a strong exothermic process, which is different from the carbon-thermal reduction reaction. The heat released by this reaction locally promotes the fusion of the sample. According to Figure 9, the disappearance of the interface between the particles in the sample starts at 800 °C, and the sintering phenomenon occurs in the same neighborhood. As shown in Figure 5B4, the whole sample melted after further increases in temperature.

$$ZnO(s) + Fe(s) = Zn(g) + FeO(s) \Delta G_{10}^{\theta} = 201066.8 - 138.33T \tag{12}$$

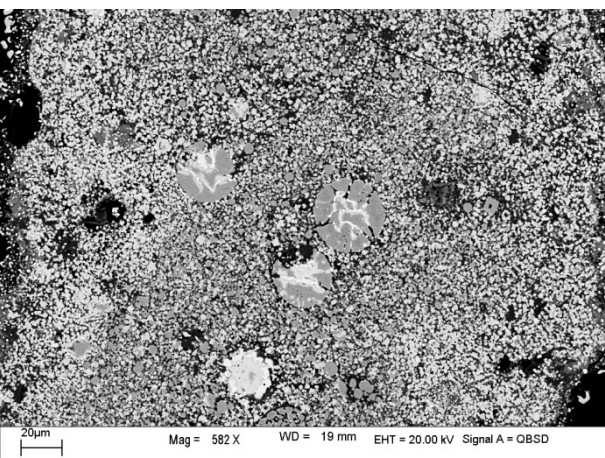

**Figure 9.** The microstructure of samples in B2 reduced at 800 °C.

In this way, ZnO should be successfully reduced to elemental zinc, and good conditions for the internal structure of the sample are required for the evaporation of zinc in EAFD. In the initial reaction, the contact is good between the EAFD and ferrosilicon alloy, and a large reaction interface exists. In this way, the interfacial reduction reaction can be carried out smoothly. When the temperature is lower than 800 °C, the reaction rate of ZnO in the EAFD reduced by silicon is slow, and the reaction is carried out relatively slowly. However, the instantaneous heat released by the reduction reaction of ferrosilicon changes the internal structure of the sample with an increase in temperature. The reduction reaction can occur smoothly, and a good condition for the escape of Zn (g) still remains. Therefore, the evaporation rate of zinc is 35.26% at 800 °C, far higher than the 0.50% of the reduction of EAFD by carbon powder. However, $SiO_2$, the oxidation product of silicon, gradually increased and a silicate-phase layer formed during the preceding reaction, as shown in Figure 5. The ferrosilicon and EAFD particles are separated by this product layer, so further reduction is limited. At the same time, the samples started to be sintered and melted, and the porosity gradually decreased, which was not conducive to the diffusion and the evaporation of the reduction product (zinc). The evaporation rate of zinc in sample B6 is 90.47%, which is less than that of sample A6, 99.80%. According to the test results, even though the reactants diffused very slowly through the solid product layer, the reduction and evaporation of zinc still occurred slowly and mainly depended on the reduction of the iron particles, as shown in Equation (12) [19,20]. On the basis of Figure 4 and Table 3, the relationship between the metal iron and FeO content suggests that ZnO is reduced by part of the metallic iron that is reduced via silicon during the heating process.

## 5. Conclusions

(1)  When carbon is used as the reducing agent, the evaporation of zinc mainly occurs in the temperature range of 900–1100 °C. For the ferrosilicon reduction, the evaporation rate of zinc increased from 700 to 1200 °C and is only 90.47% at 1200 °C. The residual zinc in the samples mainly exist in the form of a spinel solution at 1200 °C.

(2)  The beginning temperature of the metallization of the iron is 900 °C, and the percentage of the metallization is 78.67% at 1200 °C for carbon reduction. When ferrosilicon is used as the reducing agent, the variation of metallic iron content in the sample is complicated with the temperature rising due to the reaction between zinc oxide and metallic iron at a high temperature.

(3)  The microstructure of the samples changed with an increase in temperature. When the carbon is the reducing agent, the relatively independent state among the particles is still preserved in the sample at 1100 °C, which provides a good condition for the diffusion of CO and zinc (g). The sinter of the samples starts locally at a relatively low temperature, and the diffusion of Zn (g) is hindered during the ferrosilicon-reduction process.

**Author Contributions:** M.Z. drafted the manuscript and conducted the experiments, Q.M. and Q.Z. helped to carry out the experimental work, and J.L. analyzed the experimental data and modified and polished the draft.

**Funding:** The work was supported by the National Natural Science Foundation of China (No. 51974210), the Hubei Provincial Natural Science Foundation (No. 2019CFB697), the China Postdoctoral Science Foundation (No. 2014M562073), and the State Key Laboratory of Refractories and Metallurgy.

**Acknowledgments:** The authors would like to thank the anonymous reviewers for their constructive comments and suggestions.

**Conflicts of Interest:** The authors declare no conflicts of interest.

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
