# Peer review of "An Experimental Study on the Reduction Behavior of Dust Generated from Electric Arc Furnace"

_applsci, doi:10.3390/app9173604_

Round 1

Reviewer 1 Report

The manuscript can be published in its current state. The reviewer's suggestions have been included.

Author Response

Dear reviewer,

  Thank you very much for your useful comments and suggestions on our manuscript. We have modified the manuscript accordingly. The detailed corrections are seen in the attachment.

Best regards.

Mengxu Zhang, Jianli Li, Qiang Zeng, and Qiqiang Mou

Reviewer 2 Report

The article entitled "Experimental study on reduction behavior of dust generated from electric arc furnace" is an interesting manuscript about the utilization of carbon and ferrosilicon to treat EAFD although this topic was widely studied by several authors (not a lot information is provided in the introduction about other researches in this line, please improve the bibliography). The language of the manuscript is many times confussing and difficult to follow, so the English of the manuscript must be improved before considering the manuscript for publication.

Some other questions: why red paragraphs can be found in the document?

Abstract: the 99.8% is obtained at what temperature?

The first sentence (starting in line 29): 1-2% refers to what?

Sentence starting in line 33: apart from the low zinc, iron, lead, etc. Are not problematic?

Line 51: OxyCup process product is the molten iron. Whtado you mean?

Line 97: why the crucible must be dried?

Line 111 and line 123: CaO increases or not?

Line 126: 2.05 wt. Zn seems a bit high for any product to be used in the ironmaking, explain the uses of the final products? How zinc is collected?

Line 219: Boudouard

Author Response

(The authors gave the same response as above.)

Reviewer 3 Report

The article is well constructed, and its arguments are properly articulated and supported by evidence. I find it particularly useful to the readers of Applied Sciences. I am satisfied with the paper in its present form, and recommend it for publication to this journal.  

Author Response

(The authors gave the same response as above.)

Round 2

Reviewer 2 Report

Thank you for the answers. The manuscript is now acceptable for being published.

This manuscript is a resubmission of an earlier submission. The following is a list of the peer review reports and author responses from that submission.

Round 1

Reviewer 1 Report

The manuscript studies the reduction, in an induction furnace, of EAF duts by means of coal and ferrosilicon, explaining the mechanism of the process as a function of temperature. The undersigned may be of interest to the readers of APPLIED SCIENCES but it is necessary to clarify, complete and discuss some issues before accepting the manuscript for publication.

* The objective of the work must be clarified: enrich the dust in iron, recover the evaporated zinc or simply study the mechanism of the reduction process.

* Introduction: The introduction is very short and does not reflect the state of the art. For example, there is no reference to the Waelz zinc recovery process from steelmaking dusts, nor is reference made to studies on briquetting and recirculation of dusts in the electric furnace to increase zinc contents, improve iron metallization and reduce process costs. Authors should strive to summarize the state of the art properly.

* Table 1: The origin of the dusts studied should be clarified since the Cr2O3 content is much higher than normal. On the other hand, the authors do not refer to the contents of chlorides and fluorides. The dusts composition table must be completed.

* Line 81: 11g of ferrosilicon does not correspond to 22 wt% (line 78). The authors must clarify it.

Line 85: Induction oven. The EAF dusts is very fine and surely as a result of the induction a part of the mixture can leave the crucible. Authors should indicate how they performed the experiments to avoid this problem.

Lines 103-105: Correct the subindices

* Figure 3: Correct Y axis (Evaporation rate of Zn (%))

* Metallization is not sufficiently discussed. An additional discussion is necessary and to establish the differences between the A and B series from a physical-chemical and metallurgical point of view.

Line 154: The EDS spectrum does not appear, it may be convenient to include it.

* The XRD diagram of sample B6 (Figure 6) indicates that it is composed of ZnO, ZnFe2O4 and Fe3O4. The comments on lines 158-163 are not consistent with Fig. 6. The authors must clarify it.

* The evaporated Zn has been collected ?. All thermodynamic equations raised in the work are based on obtaining Zn (g) however, Zn (g) oxidizes rapidly to form ZnO (s). Authors should clarify this aspect.